# Negshell casting: 3D-printed structured and sacrificial cores for soft robot fabrication

**Pornthep Preechayasomboon**[1]*, **Eric Rombokas**[1,2]

**1** Mechanical Engineering, University of Washington, Seattle, WA, United States of America, **2** VA Center for Limb Loss and MoBility (CLiMB) VA Puget Sound Health Care System, Seattle, WA, United States of America

* prnthp@uw.edu

**Data Availability Statement:** All data and files related to the manuscript is available at https://github.com/negshell/negshell.github.io/ and

## Abstract

Soft robot fabrication by casting liquid elastomer often requires multiple steps of casting or skillful manual labor. We present a novel soft robotic fabrication technique: negshell casting (negative-space eggshell casting), that reduces the steps required for fabrication by introducing 3D-printed thin-walled cores for use in casting that are meant to be left in place instead of being removed later in the fabrication process. Negshell casting consists of two types of cores: sacrificial cores (negshell cores) and structural cores. Negshell cores are designed to be broken into small pieces that have little effect on the mechanical structure of the soft robot, and can be used for creating fluidic channels and bellows for actuation. Structural cores, on the other hand, are not meant to be broken, and are for increasing the stiffness of soft robotic structures, such as endoskeletons. We describe the design and fabrication concepts for both types of cores and report the mechanical characterization of the cores embedded in silicone rubber specimens. We also present an example use-case of negshell casting for a single joint soft robotic finger, along with an experiment to demonstrate how negshell casting concepts can aid in force transmission. Finally, we present real-world usage of negshell casting in a 6 degree-of-freedom three-finger soft robotic gripper, and a demonstration of the gripper in a robotic pick-and-place task. A companion website with further details about fabrication (as well as an introduction to molding and casting for those who are unfamiliar with the terms), engineering file downloads, and experimental data is provided at https://negshell.github.io/.

## Introduction

The fabrication of soft robotic structures is often a tedious process with multiple steps [1–3]. Soft robots benefit greatly from complex mechanical structures and geometric features to achieve actuation and sensation. Unfortunately, the soft and compliant nature of the materials used for soft robots, such as silicone or urethane-based elastomer, are only compatible with few fabrication methods such as injection molding and casting. Casting has tremendous benefits in terms of the precision in the external surface of the casted part, which is why it is often used for prosthetics and props for the film production industry, and for end-user products in the plastic molding industry. However, soft robots have structures that span both the external

additional details for methods are available at
https://negshell.github.io.

**Funding:** The author(s) received no specific
funding for this work.

**Competing interests:** The authors have declared
that no competing interests exist.

and internal space and most rely on internal features to actuate and sense. For example, the ubiquitous PneuNets [4] actuator relies on expanding internal bladders, while some sensing modalities require fluids embedded within the soft robot [5]. Casting these internal features is not a straightforward process. The fabrication process often requires multiple steps: first, a "core" that will create the internal features must be casted around and removed, then, another section is additionally casted to complete the body or another piece of the body is bonded onto the first part. Because these cores have to be removed during the fabrication process, they either: 1. have geometry constraints, such as minimal undercuts and overhangs, to prevent difficulty of removal, or 2. are made to be dissolved later, also known as lost-wax casting [3]. Lost-wax casting enables more complex geometry for the core, but the core itself must be cast from wax, which is prone to shrinkage and breakage. Aside from wax, 3D-printable dissolvable materials can be used to create arbitrarily-shaped molds and cores, such as polyvinyl alcohol (PVA) which dissolves in hot water or acrylonitrile butadiene styrene (ABS) which dissolves in acetone [6]. However, dissolving the material takes a large amount of time (more than 3 hours, according to [6]) and requires the core to be accessible from the outside of the soft robot to drive liquid solvent through.

3D printing soft robots is another heavily-researched fabrication process. 3D printing enables internal and external features to be fabricated directly and with multiple materials simultaneously. However, some solutions are prohibitively slow for mass manufacturing while others have limitations in geometry due to the absence of support material [7, 8]. Resin-jetting, multi-material printers such as the Objet Connex (Stratasys, Ltd.) series have both support material and elastomeric-like resins, but suffer from poor mechanical properties when compared to typical silicone rubbers [9]. Some stereolithography (SLA) [10, 11], digital light projection (DLP) [12–14] or continuous liquid interface production (CLIP) printers have high-resolution and fast print times, but can only print with one material at a time, which leads to support structures that need to be printed from the same material and then later removed. Enclosed chamber-like features, such as for bellows and deformation sensors, printed with SLA-style printers also suffer from cupping artefacts which can lead to ruptures [15].

High resolution 3D printing with SLA-like methods does have its benefits, however, in the casting process, as the features in the outer shell molds can be as complex and intricate as the designer desires due to SLA's high-resolution. Furthermore, resin based printing can create thin-walled parts and small cavities for use in microfluidic devices [16]. In this paper, we combine the benefits of casting liquid silicone rubber and SLA-based resin printing by 3D printing thin-walled cores for the internal features of soft robots that are meant to be left in place after casting. This greatly reduces the steps required for fabrication as casting is only done once—since the core is never removed. These thin-walled cores can be used as sacrificial elements such as for creating expanding bellow chambers and fluidic channels, or used as passive structural elements such as fingernails or bony features in soft robotic fingers. Another benefit is that since the the thin-walled cores are devoid of material, they can occupy space in the form of air instead of heavy silicone rubber, which is useful for creating features that require high stiffness and low weight. We characterize these sacrificial thin-walled cores (negative-space eggshells, or *negshells*) and non sacrificial cores (structural cores) casted in silicone rubber through mechanical testing. We also provide the fabrication steps for 3D printing our cores and molds, and guidelines for designing soft robots that employ our fabrication method along with an example to show the efficacy of the cores when combined. We also present an example application in the form of a three-finger gripper, as shown in Fig 1, performing simple pick-and-place tasks while mounted on a robotic arm.

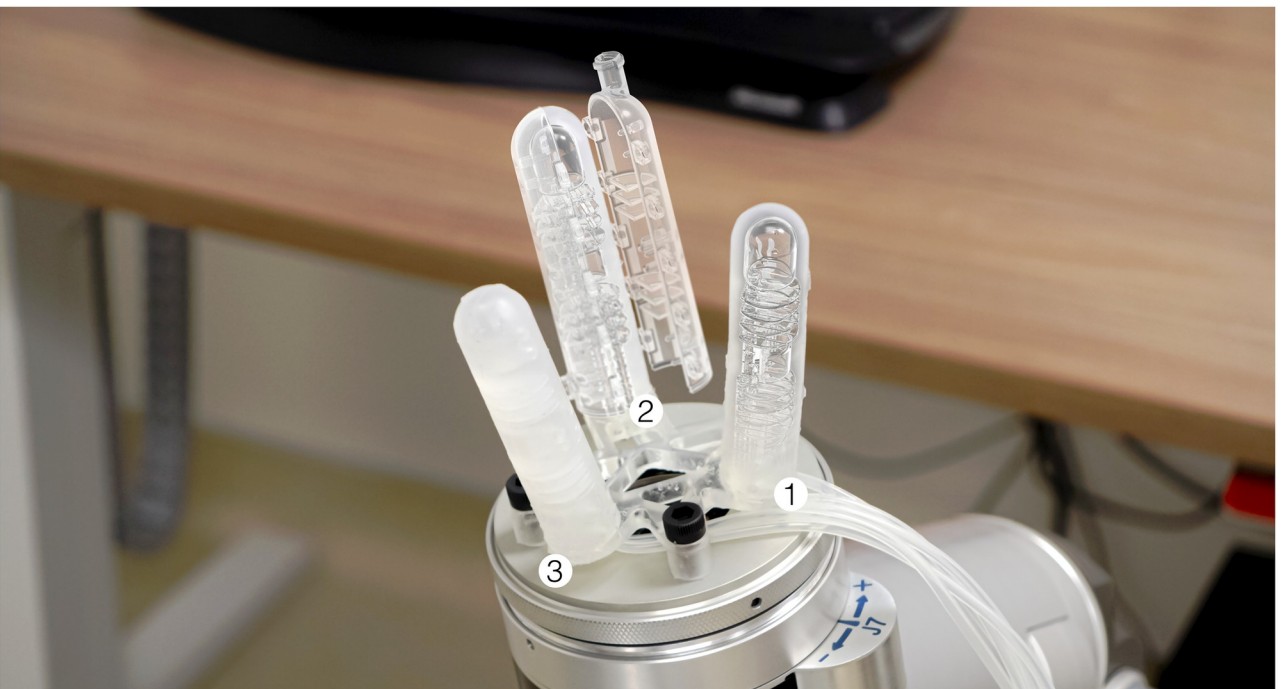

**Fig 1. Negshell casting used for a three-finger soft robotic gripper.** Pictured are three stages of negshell casting from the rightmost to the leftmost finger: 1. 3D-printed structural and negshell cores, 2. The cores placed in 3D-printed molds to be injected with silicone rubber, and 3. A completed finger ready to be actuated.

## Design and fabrication concepts

In this section, we describe the design of each component for negshell fabrication. We contextualize and describe sacrificial and structural cores. Next, we describe how these cores interact with the remainder of the mold and how they are affixed into place for casting. We then present the design of an example application, a bellow-jointed finger incorporating both kinds of cores, and demonstrating how this relatively common design theme can be miniaturized and fabricated faster and more reliably.

### Sacrificial cores

Soft robotic features such as expanding bellows or sensor cavities require a void to be left after fabrication that is later filled with fluids. As described previously, these voids are often created by solid geometry (cores) that represents the void that is later removed either by means of dissolving (lost-wax casting) or removed during the casting process. In our fabrication process, we replace these cores with 3D printed thin-walled volumes (negative-space eggshells or negshells) that are meant to be left in place or broken into small pieces that minimally mechanically effect the surrounding structure. The negshell cores are simply the outer surface of the desired core with a thickness of 0.4 mm, which is the minimum supported wall thickness for the Formlabs' resin we use to print them. This minimum thickness is used to maintain the shape of the volume during handling while being fragile enough to break under minimal force. To help promote breakage, a cross-hatch pattern, shown in Fig 2, is cut throughout the surface, as shown in Fig 3. The patterned minuscule slots also serve as channels for uncured resin to escape during the print process. Finally, 1.5 mm diameter holes are created for support

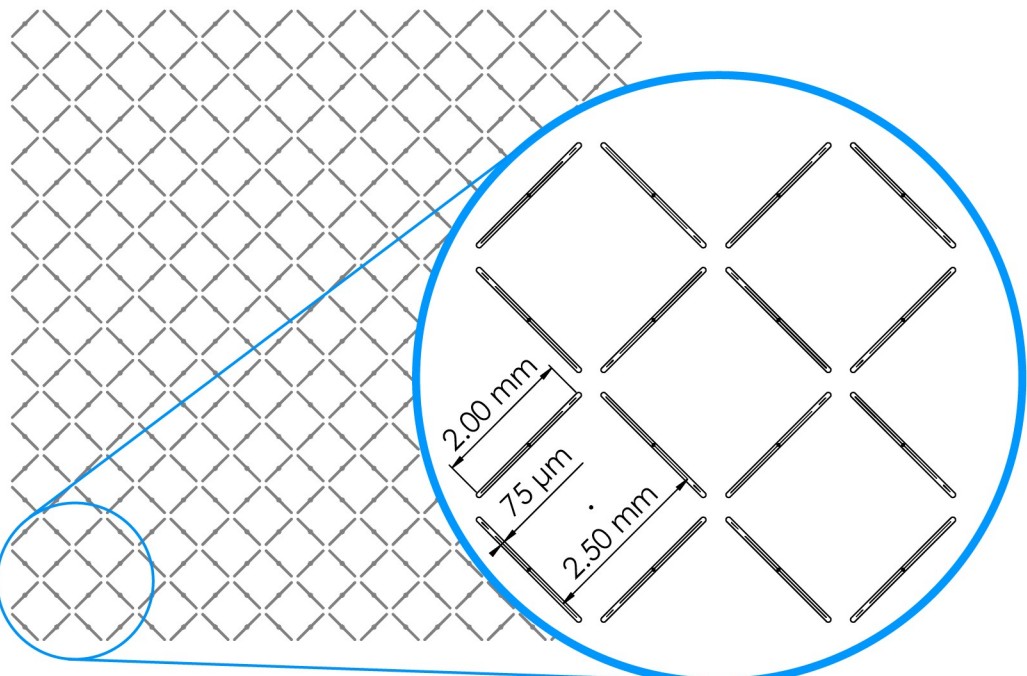

**Fig 2. Cross-hatch pattern for perforating negshell cores.** The cross-hatch are 75 wide by 2 mm long slots oriented in a 2.5 mm by 2.5 mm square pattern. The slots are designed to be small enough to be partially fused during printing to keep geometric integrity but also become fragile under load.

structures for suspending the cores in the mold, which is further explained in the **Outer molds and support columns** section.

We design our negshell cores as parts in SolidWorks, but the process can be adapted to any modern CAD software. We start by modelling the solid representation of the desired core. Then, the core is hollowed out first by copying the entire external surface followed by thickening it to 0.4 mm inwards. Finally, the hollow shell is perforated using the cut-extrude command with the previously described pattern. A brief overview of the design steps in shown in Fig 3. The part is then exported as a stereolithography (.STL) file for printing. We then use Formlabs' Preform software to prepare the negshell part for printing. The parts are first oriented for easy support removal and then Preform generates the necessary support structures for the parts. We chose a support density of 0.5 and a support touch size of 0.4 mm. Our parts were printed using Formlabs' Clear V4 resin with a layer size of 100. Additional details for fabrication, sample parts as well as .form files for Preform are available at our companion website: https://negshell.github.io/.

Due to the SLA printing process, the parts will be covered with uncured resin after printing. The residual resin must be removed prior to usage. Typically, parts are cleaned by submerging them in 95% isopropanol (IPA) for a 5 to 15 minutes and scrubbed gently with towels or brushes. However, we found that submerging negshell parts in IPA can cause them to swell and crack prematurely due to the absorption of IPA. Instead, we use a paper towel doused with IPA to wipe away the residual resin and the parts are immediately dried with a dry paper towel. The parts can also be further cured using ultraviolet (UV) light in a UV enclosure to attain higher strength and cure any remaining resin. Finally, the support structures are removed using flush-cutters.

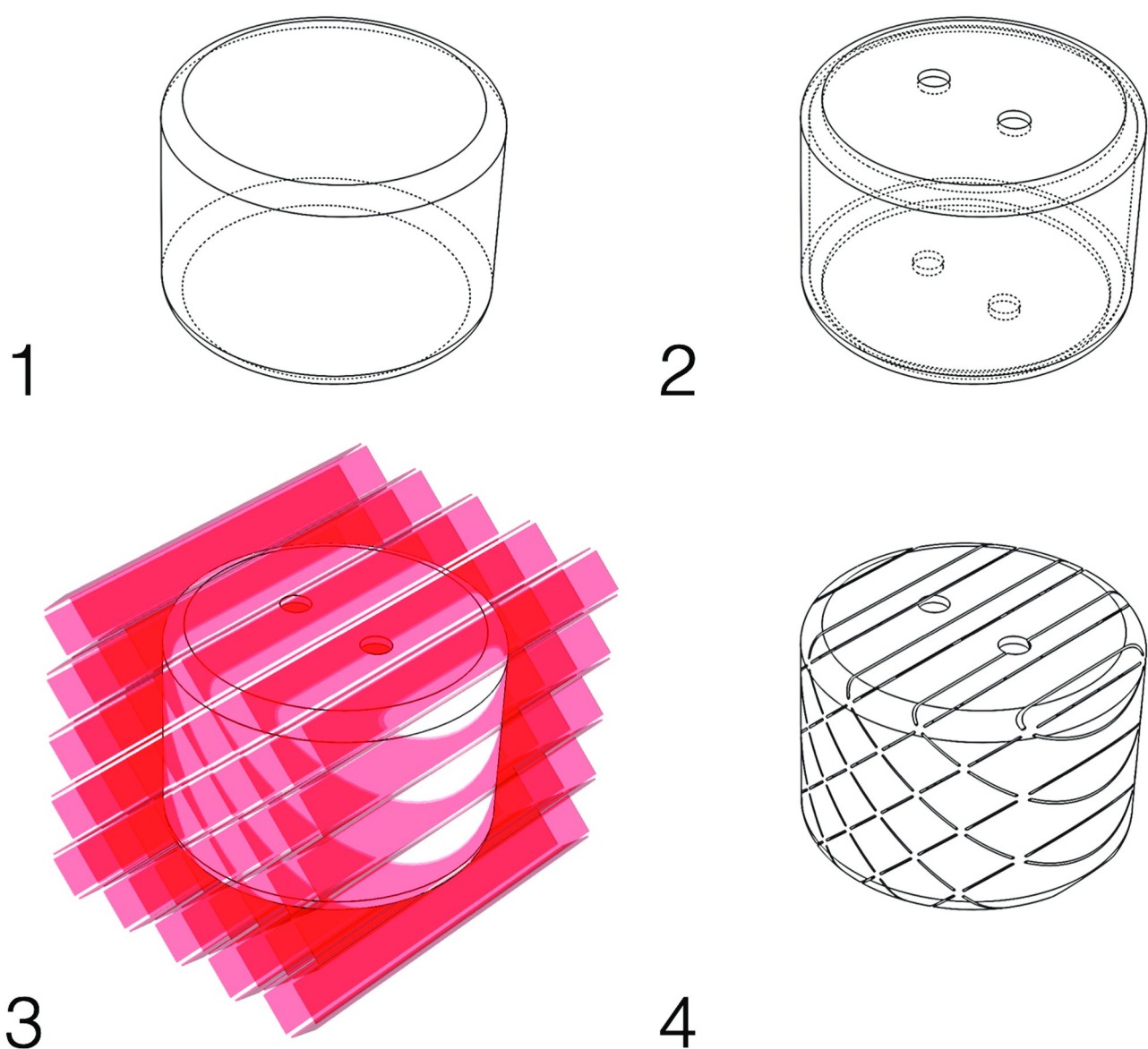

**Fig 3. Creating a negshell core in CAD.** 1: The solid model representing the core. 2: The core is hollowed to a 0.4 mm shell. 3: Cutting the cross-hatch pattern into the hollowed core. 4. The final negshell core ready for 3D printing.

## Structural cores

Casting is often performed by injecting or pouring a homogeneous material and having it set. Casting multiple materials is possible, but requires a multiple-step process, such as overmolding: casting additional elastomer on top of an existing injection-molded rigid part in a separate mold. Overmolding is often used for electric power tools and cable assemblies to add compliance to an otherwise rigid part. For soft robots, having the entire structure comprised of a single material limits the design space due to the high compliance that is homogeneous throughout. By embedding semi-rigid, passive structures inside the soft robots' body, the localized mechanical properties at those structures can be tuned. We achieve this by utilizing cores similar to negshell cores but do not have the perforations. Such structures provide a semi-rigid internal skeleton that provides a stiffer backing to exert force akin to fingernails in humans, or

provide lightweight void space for areas that connect one structure to another that would otherwise be heavy and undesirably non-rigid if made entirely from elastomer.

We design and build these structural cores in the same manner as the negshell cores: a hollow shell with minimum wall thickness. The thickness can be tuned to create varying stiffness in the resulting part. We present a characterization of these parameters in **Characterization** section.

## Outer molds and support columns

The cores must be suspended in the outer mold at its intended location during the casting process, as shown in Fig 4. It should be noted that this is a limitation of this fabrication method as the structure necessary to suspend the cores prevents the internal features from being completely sealed. However, the support structures can be made small in size and the resulting holes can be sealed with a small amount of adhesive or filler, such as Sil-Poxy (Smooth-On, Inc.).

The support structures (or standoffs) are round 2.5 mm columns that reach from the inner surface of the outer mold up to the core. At the end of the column, there is a smaller column with a beveled tip approximately 1.5 mm in diameter that will penetrate and secure the corresponding hole on the core, as shown in Fig 4. Depending on the geometry of the core, the support columns can be placed at multiple locations around the mold and core. To aid in the ease of removal of the outer mold after casting, the columns should be oriented in the same pulling direction when removing the mold, although this design rule can be somewhat relaxed due to the compliance of the elastomer.

The outer mold can be as simple as two halves of a shell that defines the external features of the desired part, or a complex multiple part mold to accommodate undercuts or overhangs and features that would be otherwise impossible to be demolded with a two-part mold. As our focus of this paper is on the internal features, we designed most of our parts to be cast in simple two-part molds. The molds are often split at the central plane of the desired part that creates

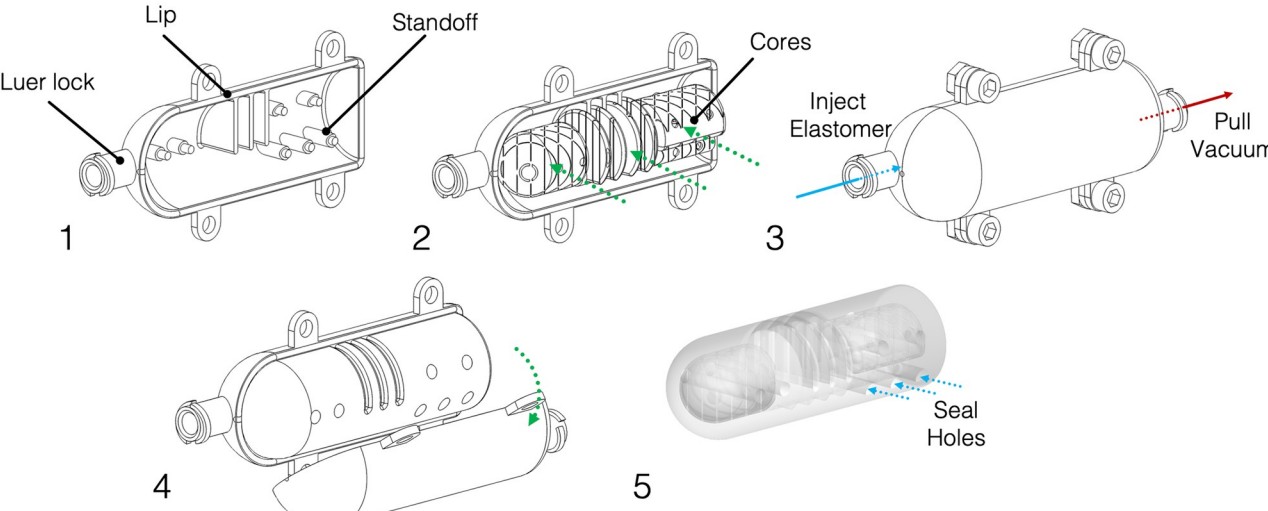

**Fig 4. General fabrication steps for negshell casting.** 1: The mold is prepared. The Luer lock, lip and standoff features are highlighted. 2: The cores are inserted into one side of the mold. 3: The mold is clamped shut with fasteners and elastomer is injected via a syringe. Another syringe can be used at the other end to pull a vacuum to help with liquid elastomer flow. 4: Once the elastomer has set, the mold haves can be removed. 5: The completed soft robot component.

the least undercuts. In this paper, our molds have the following set of features to aid in fabrication: 1. The seam connecting the two halves of the mold have a lip to help with alignment and prevent liquid elastomer from escaping. 2. There are 3.1 mm holes intended for M3 screws and nuts along the perimeter of the mold to clamp the two halves together. 3. Instead of pouring liquid elastomer in, we opt for injecting the liquid elastomer using a syringe. We add two Luer lock ports (Fig 4) at opposite ends of the mold: one for injecting the liquid elastomer with a syringe, while the other port is used for drawing a vacuum to assist in flow and for increasing pressure to eliminate air bubbles.

Our molds are 1.5 mm thick and are printed using the same Clear V4 resin as the negshell cores. The molds are thoroughly washed with IPA to minimize any residual resin, as uncured resin can inhibit curing of the Platinum-cure silicone that we use.

## Design of a bellow-jointed finger

Here we present an example of a soft robotic finger that utilizes both the structural cores and negshell cores. The finger has multiple bellows connected to an inlet tube that is the result of a negshell core. The tip and base have stiffening and weight reducing elements created by structural cores, as shown in Fig 5. The completed finger is a cylinder approximately 18 mm in diameter and 55 mm in length with a domed tip and weighs approximately 15 grams. The bellow negshell core and accompanying mold features create 1 mm wide internal bellows with 1 mm thin walls. The structural cores are located approximately 2 to 3 mm deep from the surface of the finger.

To fabricate the finger, the cores and molds were first 3D printed from Clear resin with the same parameters as mentioned previously. Once the mold and cores have been cleaned from resin residue, the mold's inner surface was lightly brushed with Pol-Ease 2500 (Polytek Development Corp.) release agent and the cores were placed in one half of the mold. The second half was then clamped shut with M3 screws and nuts. 15 grams of liquid silicone Plat-Sil Gel 25 (Polytek Development Corp.) was prepared by vigorously mixing equal amounts of part A and part B which was followed by vacuum degassing to remove trapped gasses, as we found

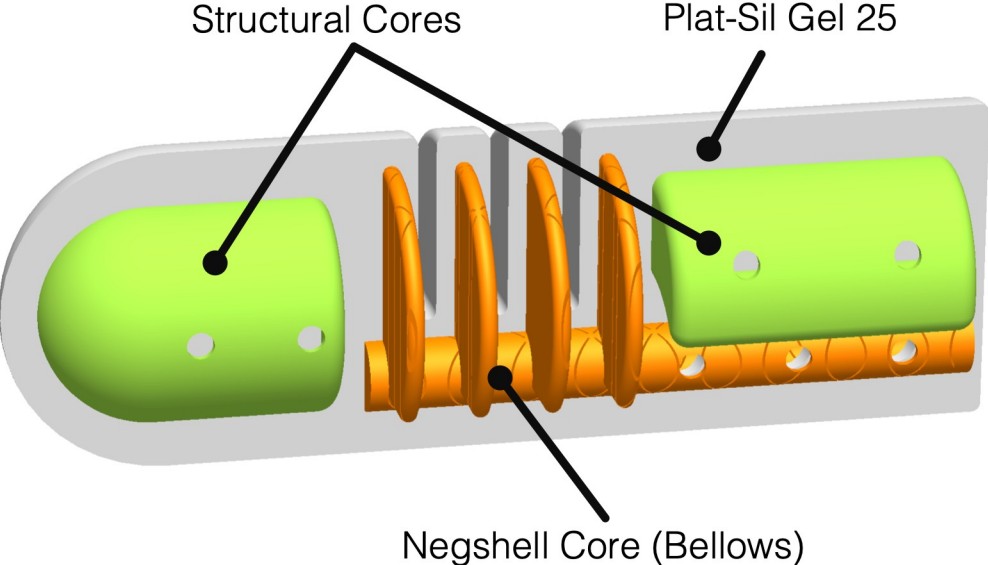

**Fig 5. Cross-section of soft robotic finger.** A 3D CAD model of the soft robotic finger is cut longitudinally to show the locations of the non-sacrificial cores and the negshell core that create the bellow.

that air bubbles can lead to leakages near the bellows due to the low thickness. The liquid silicone was then poured into a 30 mL syringe and the silicone was injected into the mold via the Luer lock port at the tip. The Luer lock port at the other end serves as a vent but can also be used with another empty syringe to draw a vacuum to aid in the flow of liquid silicone. Both syringes were left in place during curing to prevent silicone from flowing out of the mold. After waiting for the silicone to cure, which is approximately one hour for Plat-Sil Gel 25, the syringes was removed and the mold was opened. Small holes created by the standoffs were sealed with Sil-Poxy (Smooth-on, Inc.) silicone adhesive. The same adhesive was used to bond a 4 mm silicone tube to the inlet. Finally the negshell cores were crushed by hand, completing the finger.

During operation, as pressure is increased in the cavity created by the negshell cores, the bellows expand and push against each other which causes the finger to bend. Since the silicone and 3D printed cores do not form a bond, the fluid used to actuate the finger can travel along the surface of the bellow fins and gaps left after crushing the negshell core walls. The bellows also expand against the stiff structural cores, creating a clear path for force transmission from the tip of the finger to the base of the fingers.

## Characterization

To evaluate the efficacy of the cores, the following section presents several mechanical characterizations of both negshell and structural cores. We characterize: 1. the force required to break the negshell cores, 2. the stiffness of elastomer with negshell cores post-break, 3. the stiffness of elastomer with structural cores, and 4. a comparison of the blocked force output of bellow-jointed fingers with and without structural cores.

### Specimens and experimental setup

Specimens for both types of cores are hollow cylinders with a diameter of 16 mm and a height of 11 mm. The cores were then casted into a cylinder 20 mm in diameter and 15 mm in height with Plat-Sil Gel 25 Platinum-cure silicone, which resulted in an elastomer thickness of 2 mm across the whole specimen. The casting process is similar to the processes detailed previously. We fabricated three samples of each type of core, including comparable samples that can be achieved with traditional casting methods, as shown in Figs 6 and 7. All samples were tested on a TA ElectroForce 3200 Mechanical Tester (TA Instruments) fitted with a 50 mm platen and 120 N capacity load cell, as shown in Fig 8.

Each specimen was also weighed. The average weight of each specimen type from Fig 9 shows that for negshell specimens, the negshell cores contribute approximately 27% of additional weight and the structural cores reduce weight by up to 67% for these specimens.

### Characterization 1: Force required to break negshell cores

As the negshell cores are intended to break into small pieces, we characterize the amount of force required to break the negshell cores. The characterization is done by placing the negshell cores on its side and subjecting the cores to a compression load on the TA Electroforce 2300. The specimens were preloaded to deform slightly, compressed 4 mm further at 0.25 mm/s, and then released at 1 mm/s. We tested three different specimen types with the same wall thickness of 0.4 mm: 1. cores with perforations cut from the circular base of the cylinder (top), 2. cores with perforations cut from one side, and 3. cores with perforations cut at the two sides at right angles, as shown in Fig 6. The results show that the cores start to break from 20—35 N of force and had a wide range of variance between each sample, as shown in Fig 10. This is,

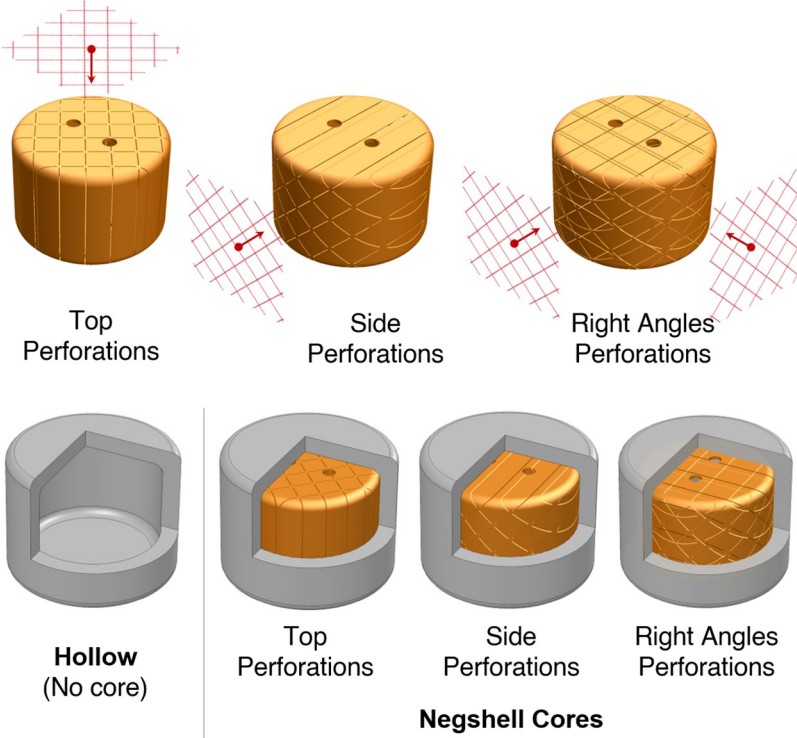

**Fig 6. Negshell core specimens.** The four types of negshell core silicone specimens tested. A zonal cross-section shows the internal features of the specimens. The hollow specimen does not contain any core. The negshell core specimens contain the three different cores with different perforation directions.

however, well within range of the force a human can exert with a pinch grasp—and is also demonstrated in S1 Video.

## Characterization 2: Stiffness of elastomer with negshell cores post-break

Negshell cores, once broken, are intended to minimally affect the mechanical properties of the surrounding elastomer and provide the same function as a removed core. To demonstrate this, we cast three different types of negshell specimens (top perforation, side perforation and

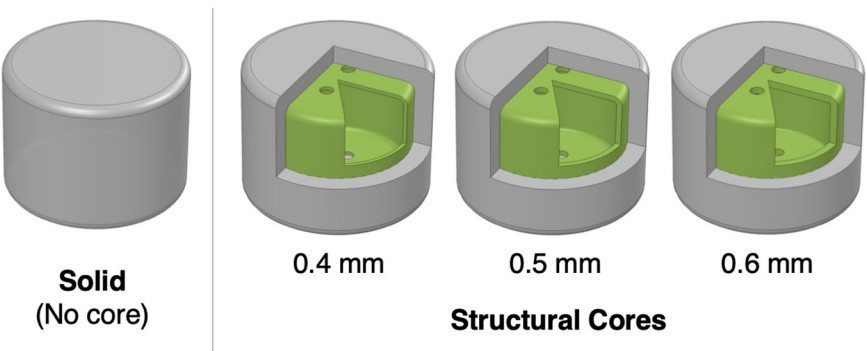

**Fig 7. Structural core specimens.** The four types of strutural core silicone specimens tested. The solid specimen has homogeneous material throughout its volume. The cores in the other specimens have a thickness of 0.4 mm, 0.5 mm, and 0.6 mm.

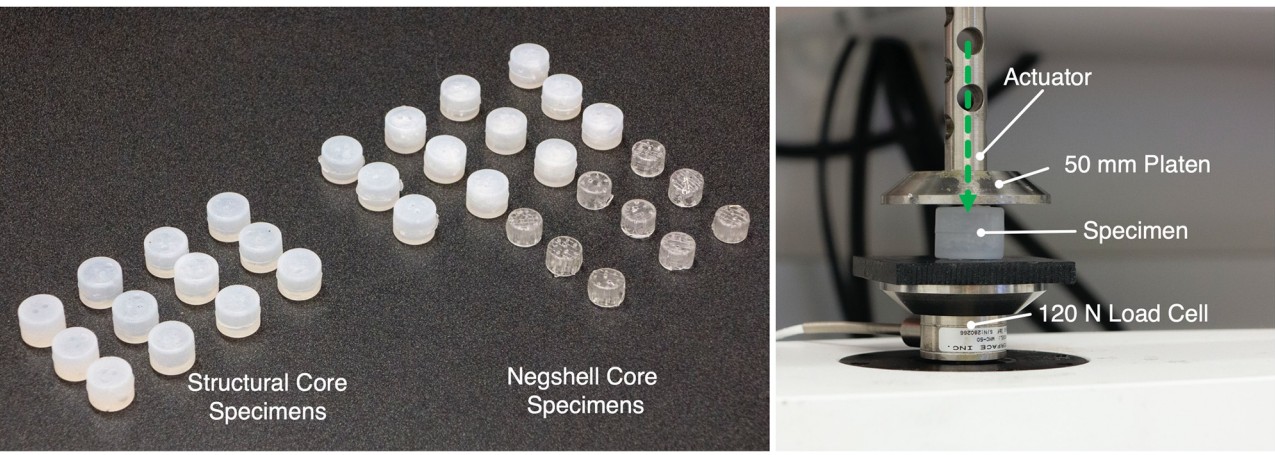

**Fig 8. Experimental setup.** (Left) All of the specimens used for testing. (Right) The specimen is placed at the center of the platen. As the actuator moves the platen down, the specimen is compressed and the resulting force is read from the load cell beneath.

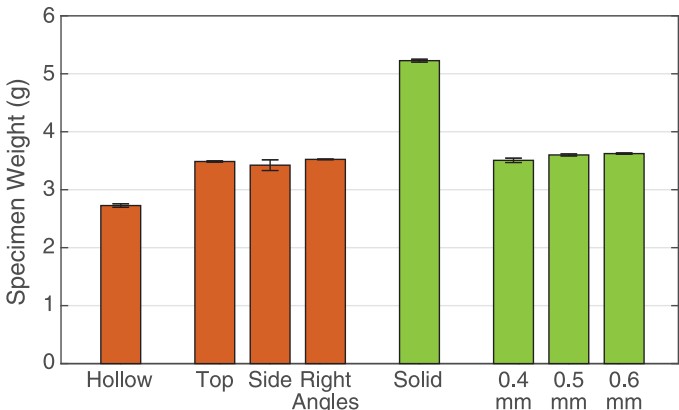

**Fig 9. Specimen weights.** Three samples of each type of specimen was weighed on a 0.01 gram resolution scale and averaged. The errorbars represent the standard deviation.

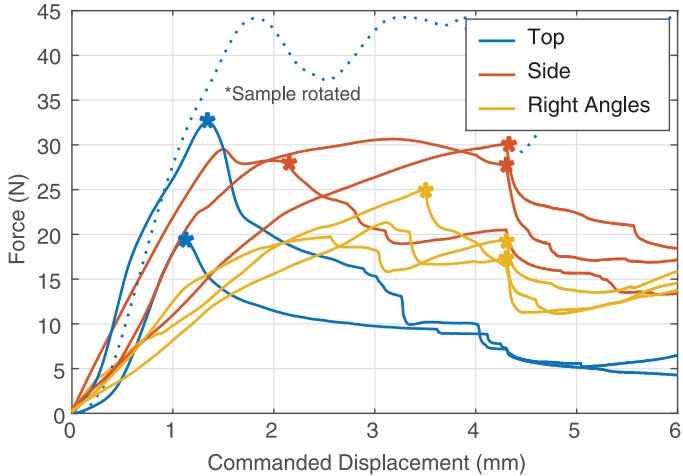

**Fig 10. Breaking force for negshell specimens.** Each specimen required different amounts of force to start rupturing, as denoted by the asterisks that precedes a sharp decline in force reading.

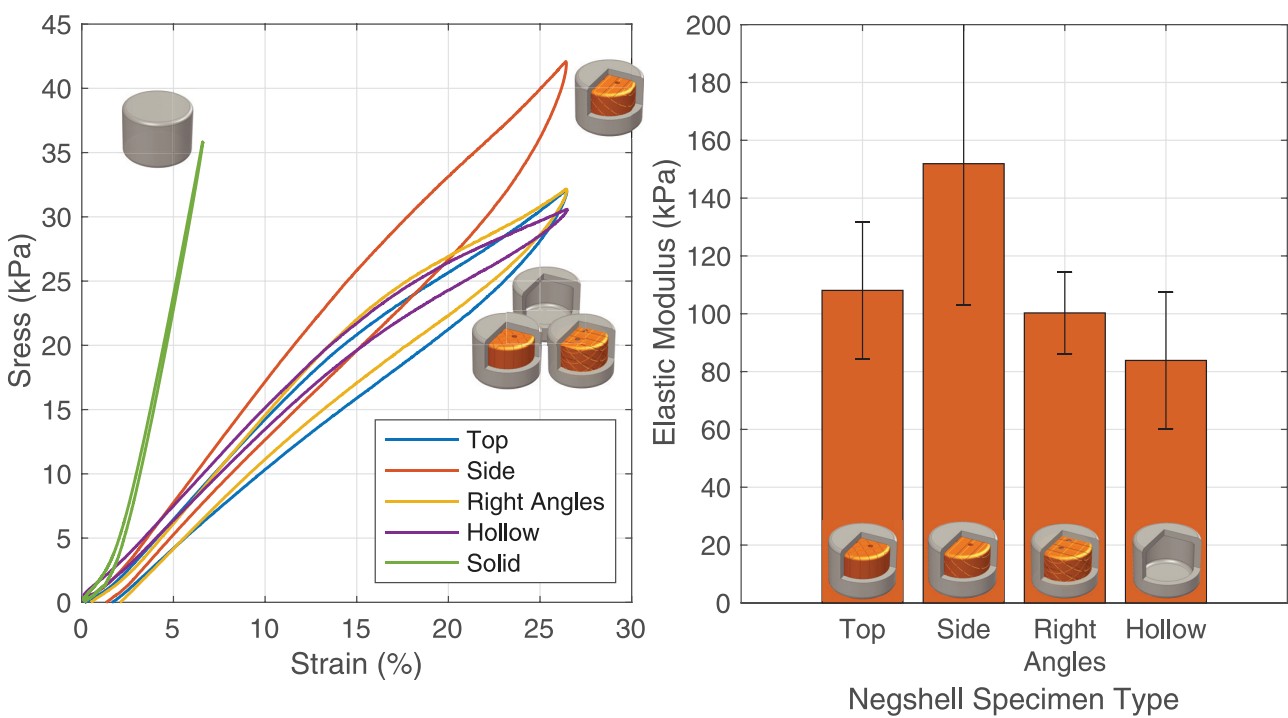

**Fig 11. Negshell cores results.** (Left) The stress-strain curve of the tested specimens show that the negshell cores minially effect the mechanical property of the specimens under compression. The stress-strain curve of a solid, homogeneous specimen is also plotted as refrence. (Right) The elastic modulus of the tested specimens.

perforation at right angles) and compare them to specimens with the equivalent resulting geometry that were fabricated with a traditional casting process: casting with a solid core in halves and then bonded together. Each specimen was first subjected to a preload of 1 mm of deformation followed by a 0.5 Hz triangle wave of compression with a peak-to-peak displacement of 4 mm for five cycles. The average stress-strain curve of each type of negshell specimen is shown in Fig 11. The stress in the specimen is calculated by dividing the force read from the load cell by the area of the base of the specimen. From the stress-strain curve, it can be implied that the Top and Right Angles specimens behave similarly to the specimens without the cores. The Side specimens have slightly higher overall stiffness, as shown in stress-strain curve and the resulting elastic modulus. The average elastic modulus (E) (derived from an approximate linear fit of the stress-strain curve) of the different core types that is shown in Fig 11 confirms that the the broken negshell cores contribute an insignificant amount of stiffness to the casted samples.

### Characterization 3: Stiffness of elastomer with structural cores

To demonstrate that structural cores with thin walls can modulate the overall stiffness of soft robotic elements, we casted specimens with structural cores with three different thicknesses: 0.4 mm, 0.5 mm and 0.6 mm and compare them to a equivalent homogeneous specimen without a core. Each specimen was subjected to a 0.5 mm displacement preload followed by a 1 mm peak-to-peak displacement triangle wave at 0.5 Hz for five cycles. We did not exceed this displacement range due to the amount of force reaching the upper limits of the load cell in some samples. The average stress-strain curve of each type of specimen and accompanying elastic modulus is shown in Fig 12. It can be seen from the resulting elastic moduli that the

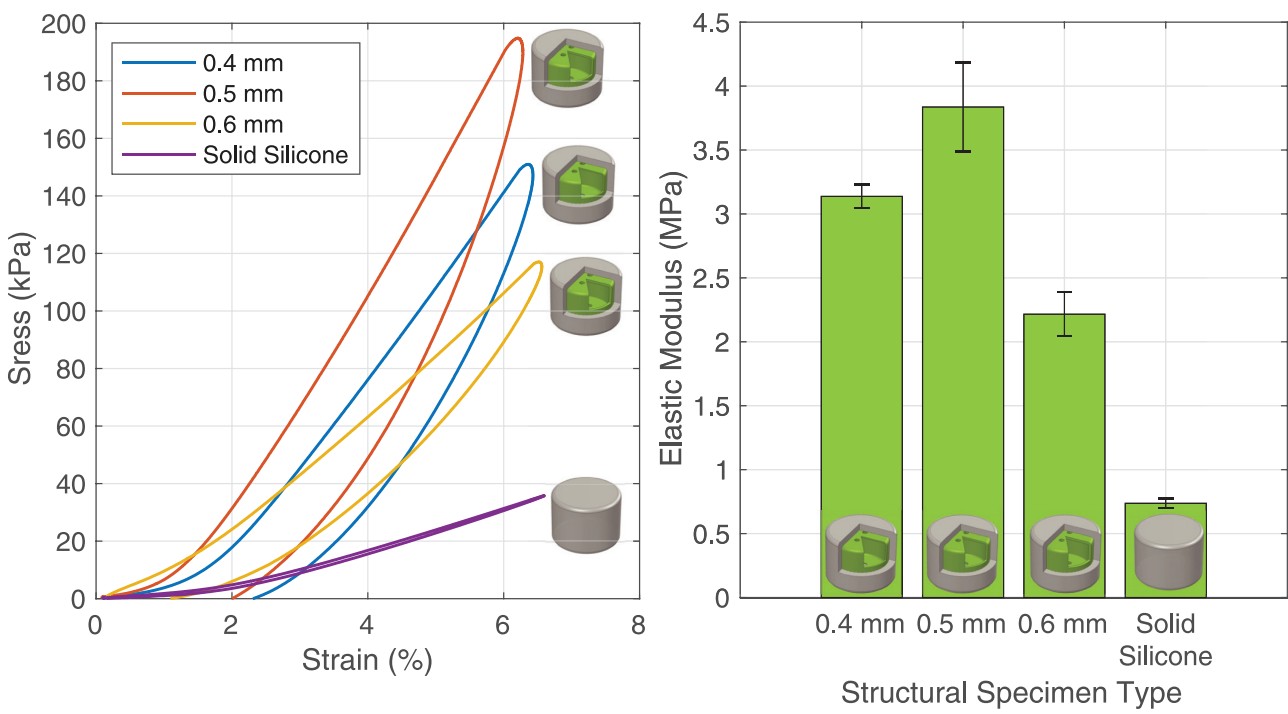

**Fig 12. Structural core results.** (Left) The stress-strain curve of the tested specimens show that the stiffness of the specimens can be greatly increased with the structural cores. (Right) The elastic modulus of the different structural cores.

stiffness of the overall structure can be increased by over 500%, while reducing the weight by 67% (Fig 9).

## Characterization 4: A comparison of the blocked force output of bellow-jointed fingers with and without structural cores

As previously hypothesized (Fig 13), the structural cores help provide a means of force transmission from the tip of the finger to the base. We test this by comparing two fingers fabricated with and without structural cores in a blocked force experiment. The finger without the structural core was fabricated by simply not inserting the structural cores into the mold. The fingers

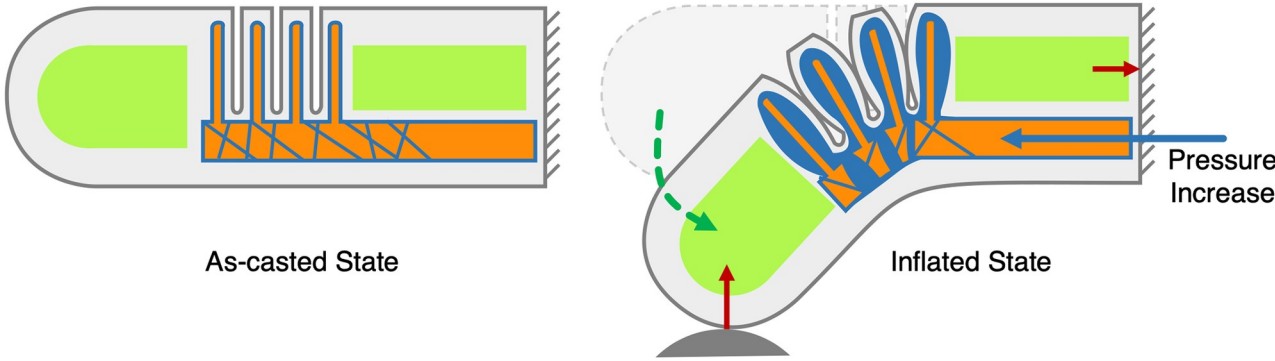

**Fig 13. Main actuation principle of the soft robotic finger.** As the actuation fluid (air/water) increases in pressure or volume, the fluid travels along the cavity left by the negshell core up to the fins of the bellows and start to expand the thin walls of the silicone elastomer surrounding the fins which creates a bending force.

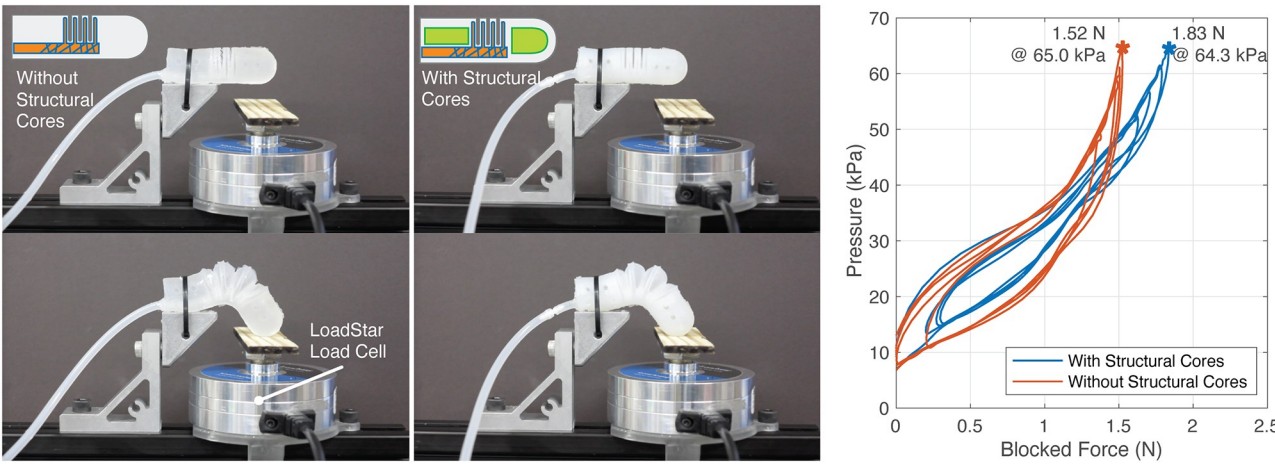

**Fig 14. Blocking force output of the bellow-jointed fingers.** (Left) The experimental setup. The fingers in the top row are in the idle state the bottom rows are at full flexion. (Right) A plot showing the relation between input pressure and the amount of force exerted by the finger.

were secured with a cable tie to a rigid aluminum structure at the base and a load cell is placed under the finger approximately at the location where the finger tip touches at 15 degrees of flexion. The finger is manually pressurized using a 30 mL syringe connected through a silicone tube to the bellow structure. An example of actuation against the load cell is shown in S2 Video.

The results from the blocked force experiment, in Fig 14, show that the finger with the structural cores can exert a maximum force of 1.83 N with 64 kPa of air pressure while the finger without the cores can only provide 1.52 N of force with the same pressure—a 20% increase in force output with the structural cores. Furthermore, the structural cores reduce the overall weight of the finger by 1.8 grams or 14%.

## Application

In this section, we combine the design elements of negshell casting in an example application where we fabricate a three-finger gripper with 2 bellow-joints per finger. The gripper demonstrates the use of negshell cores for creating expanding bellows at each joint and their fluid channels, and structural cores to aid in force transmission, local stiffness, and to secure the finger to the base. We also mount the gripper on the tool plate of a Barrett WAM 7-degree-of-freedom robotic arm (Barrett Technology, LLC) to demonstrate a simple real-world scenario of a pick-and-place task.

### Design and fabrication

Each phalanx of the gripper can be independently actuated by applying fluid pressure on each of the two bellow-joints: the proximal joint and the distal joint. The proximal joint connects the structural core in the base to the middle structural core and the distal joint connects the middle structural core to the finger tip, as shown in Fig 15. Each joint consists of four fan shaped 0.6 mm thin chambers with a diameter of approximately 16 mm, similar to the bellow-jointed finger in the previous section. A 3 mm tubular negshell core connects the bellows to the exterior of the finger through two holes in the base—one for each joint, which is later used for inserting and bonding a 3 mm silicone tubes to connect the chambers to a pressure source. The middle structural core is a half-cylinder with a thickness of 0.6 mm. The core at the fingertip is a hybrid between negshell and structural cores, where the "pulp" of the fingertip core is

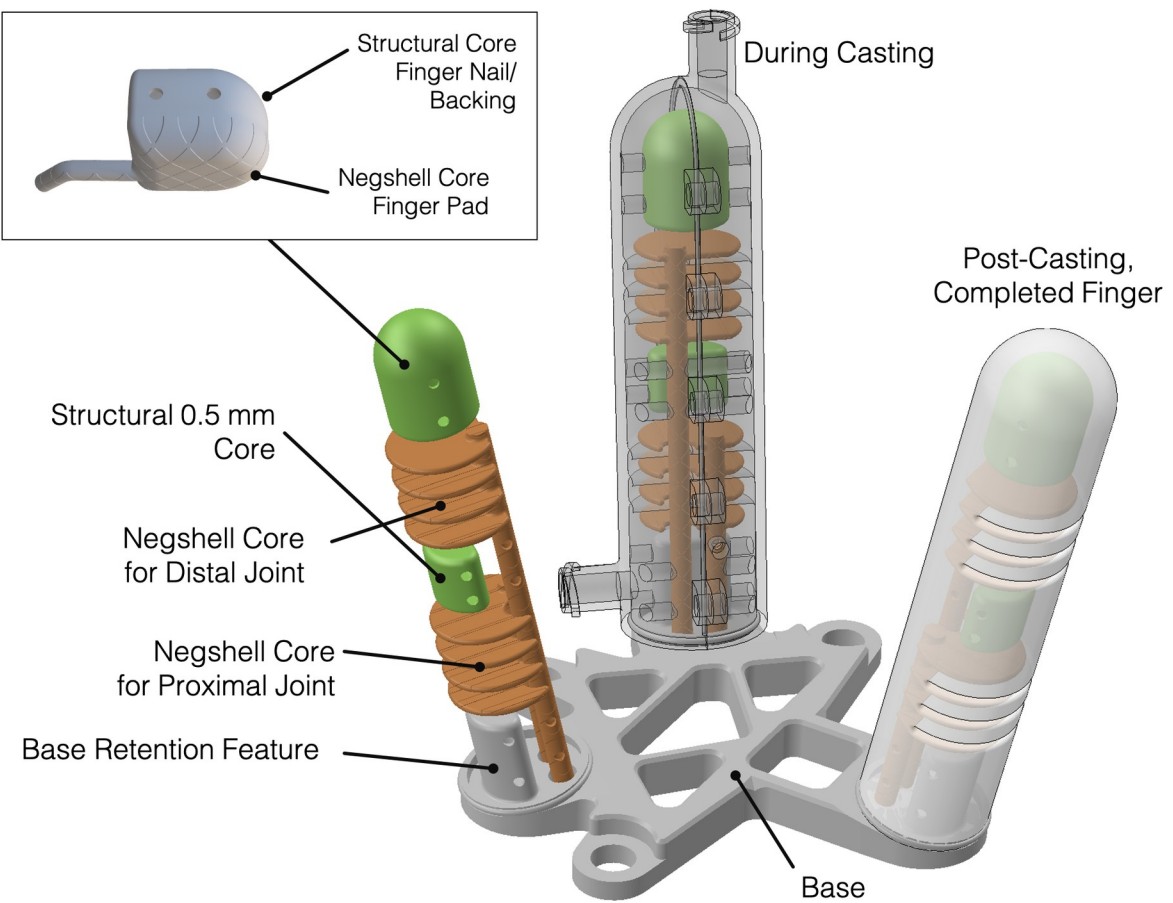

**Fig 15. Overview of three-finger gripper's fabrication.** The 3D-printed base, which serves as a part of the mold for the fingers, provides a rigid connection from the fingers to the tool plate of the WAM Arm. Two seperate negshell cores are used for each joint of the fingers. A structural core aids in rigidity and force transmission between the joints. The fingertip contains a hybrid structural-negshell core with renders a soft finger pad and solid backing, as shown in the inset. The base also contains a retention feature geometrically similar to the structural cores, where silicone will be mechanically trapped and locked in place due to the overhang.

to become soft and compliant to emulate soft human fingertip tissue while the immediate structural core connected to the negshell core emulates the hard backing provided by the bony structure and fingernails in humans, as shown in the inset in Fig 15. We intend for the fingertip to become a sensor, but it is beyond the scope of this paper. The mold is similar to the mold presented in the previous sections with one difference where the base is also used as part of the mold. Geometry similar to a structural core is integrated into the base which is used to retain the finger on the base, as shown in Fig 15.

We fabricated the gripper by first 3D printing the cores, outer molds and base using Clear resin on a Formlabs Form 3 printer. The parts were then cleaned and processed in the same manner as in the previous sections. The molds were then prepared by applying a light coat Pol-Ease 2500 to aid in demolding. One half of the mold was then assembled onto the base followed by the insertion of the various cores. The other half of the mold was then carefully closed and several M3 screws and nuts clamped the two halves shut. Approximately 30 grams of Plat-Sil Gel 25 was mixed, vacuum degassed and loaded into a syringe, then the liquid elastomer was injected into the port at the fingertip. Another syringe was used at the base to draw a vacuum or increase pressure to help with residual air bubbles. Once the elastomer set, the two

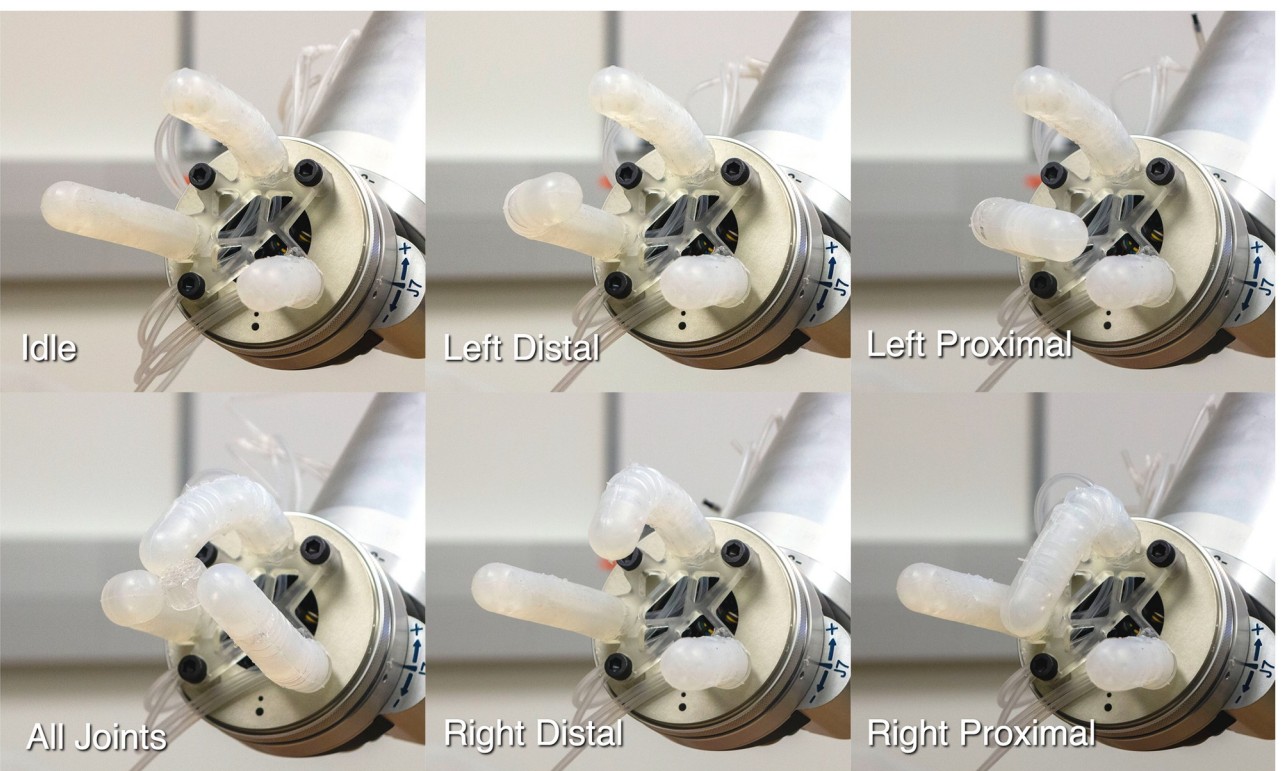

**Fig 16. Pressurization of each joint of the gripper.** (Top-leftmost) The gripper in its un-pressurized, idle state. (Bottom-leftmost) All joints being pressurized at once. (Right four photographs) A series of photographs showing each joint of two of the gripper's fingers pressurized.

halves were released and Sil-Poxy was used to seal the holes created by the standoffs, and to bond silicone tubes to the fluid channels for the bellows and fingertip. Each finger can be done one at a time, or all three at once if three sets of molds were printed. Finally, each negshell core is broken by hand and the gripper is ready to be mounted to the WAM Arm tool plate using M6 hexagon cap screws.

## Actuation

There are a total of six bellow-joints in the gripper, which can be individually actuated by modulating the pressure within the bellows of each joint. Pressure modulation can be done in multiple ways such as using multiple pulse-width modulated (PWM) valves and a pressure source [17] or multiple leadscrew driven syringes [9]. However, as robust actuation and dexterous manipulation is beyond the scope of this paper, we simply combined all the fluid channels together into a single channel so all joints are pressurized with a single pressure source, which in this case is a hand-actuated syringe. However, individual actuation of each bellow using a hand-driven syringe is shown in Fig 16 and S3 Video.

## Pick and place demonstration

As a demonstration of real-world usage of the negshell casted three-finger gripper, we performed a brief demonstration by using the gripper mounted on a Barrett WAM Arm to perform pick and place tasks of everyday items. We pre-recorded the arm movement using a "Teach and Play" program that is then played back each time the arm performs a pick and place task. The arm starts from its home position, moves to position the gripper directly above

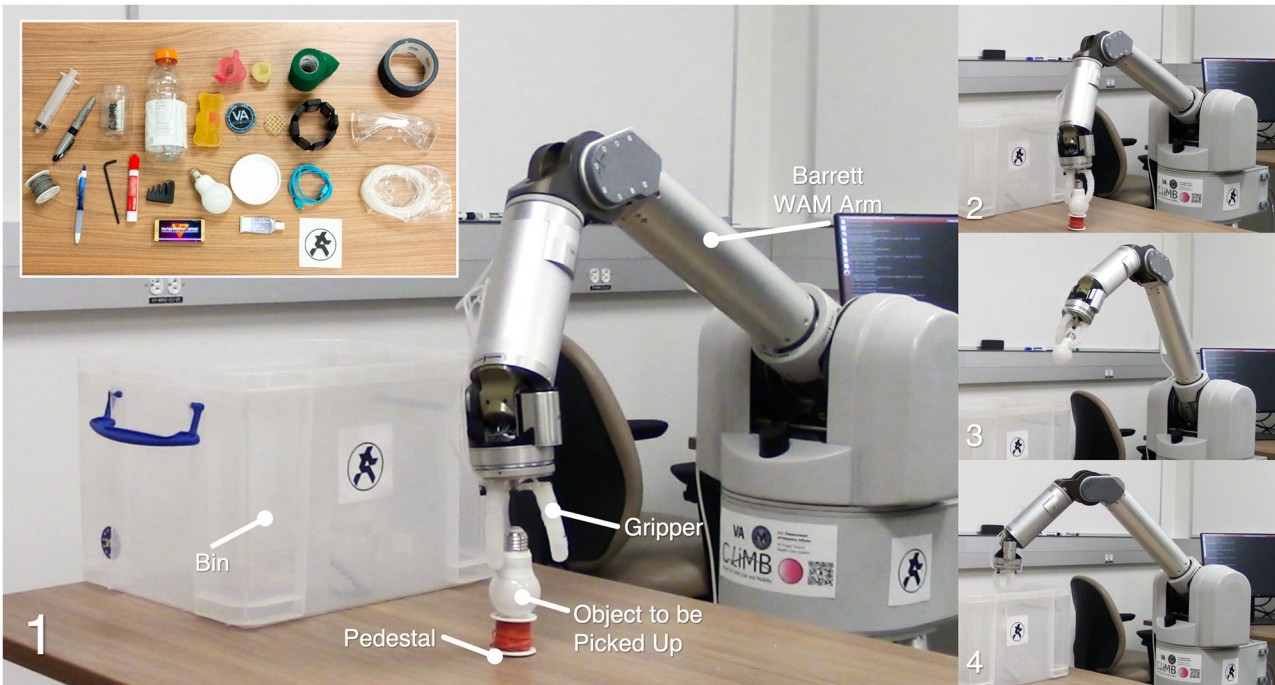

**Fig 17. Pick and place demonstration.** The three-finger gripper is shown attached to a Barrett WAM Arm. The arm goes through a pre-recorded motion while the gripper is manually actuated with a syringe (not shown). 1. The arm positions the gripper to encompass the object to be picked up, 2. All bellows in the gripper are pressurized at once using a syringe, grasping the object, 3. The arm moves the object up over the bin, 4. The object is dropped into the bin upon the release of pressure. Inset: the objects that were successfully picked and placed.

the object to grasp, and moves the gripper down to encompass the object with the fingers. The fingers are then closed using the syringe and the arm moves the object with the gripper towards a bin. Finally, the fingers are opened and the object is dropped into the bin.

We tested grasping a total of 28 objects varying in shape, size and density, with most shown in Fig 17 e.g. a water bottle, a lightbulb, and a 5 mm hex key. Some objects were handed to the robot, such as the water bottle and pen, while others were placed on a pedestal to be picked up, such as the wooden block and cloth. Sped up footage of the robot arm grasping objects can be found in S4 Video. It can be observed from the footage that the gripper successfully conforms to objects with arbitrary geometries and poses despite the bellows being actuated all at once, demonstrating proper compliance as expected from a soft robotic gripper.

## Discussion

Negshell casting opens up new possibilities for casted soft robotic designs by integrating 3D-printed parts both into the structure and the casting process—cutting down the casting process while increasing the design space significantly. The internal geometry of soft robotic structures can be much more complex than what could previously be done through casting. However, negshell casting does have limitations. Since the cores must be suspended during casting, some support structure must exist, which requires sealing the holes created by the structures. Though straightforward, sealing the holes is an additional step that is done manually and prone to human error. In the future, we hope to address this step to further streamline the fabrication process for the sake of scalability and manufacturability.

Sensing, although mentioned in the previous sections, has not been demonstrated in this paper. However, the mechanical characterization of the negshell specimens show that, under

compression, the broken negshell cores have little to no effect on the structure, thus negshell casting could possibly be used for creating sensor channels or structures. An example of this is shown in the fingertip of our three-finger gripper, where the fingerpad is deformable chamber that can be connected to a pressure sensor for simple tactile sensing. Sensing will "close the loop" for our soft robotic grippers and thus we will explore sensing and closed-loop control in the near future.

3D-printing soft robots, especially with complex internal structures, is still an active research topic. Work such as [18], still requires molds and manual labor within each layer of printing. The most notable hurdle is arguably the material science required to develop the ideal printing material. Our technique leverages already mature materials for 3D-printing and adapted it to a tried and true method for fabrication. For future work, combining the two technologies may aid in each-other's shortcomings. For instance, 3D-printing can be done on top of negshells and structural cores that are used for scaffolding which eliminates the need for sealing holes that are created during casting, while providing instant negative space that must be otherwise printed with fugitive materials.

One concern that may be raised is that the fragments that are created during fabrication may cause blockages or damage to the structure. To mitigate blockages, the design of the cross-hatch pattern may be tweaked in a manner where the distance of the perforations are smaller than the smallest channel that is exposed to the negshell core. In practice, however, we hypothesize that due to the inherent compliance of the structure, fluids would simply flow around the blockage. Furthermore, the fragments themselves are flexible and are compliant themselves due to the 0.4 mm thickness. As for damage, as crushing and breaking the negshell cores is part of the manufacturing process, the soft robotic structure has already been subjected to forces much higher than typical usage conditions. Our combination of materials shown in this paper has proved to be able to withstand the process. As for other materials, the parameters of the negshell core might require tweaking, i.e. using a thinner wall thickness or larger perforations for silicone with less hardness.

## Conclusion

We have presented a novel fabrication technique, negshell casting, that leverages SLA 3D-printing to create complex cores for use during an otherwise traditional casting processing. We present two types of cores: negshell cores and structural cores. Negshell cores are sacrificial cores that are used to create soft or flexible internal structures such as channels and bellows while structural cores are used for modulating stiffness and for weight reduction. We showed that the negshell cores have little effect on the mechanical structure of silicone elastomer and structural cores can increase the stiffness of silicone structures significantly. Our demonstration of the bellow-jointed finger and the three-finger gripper shows that negshell casting can be used for creating soft robotic actuators. In the near future, we will explore how negshell casting can enable more sophisticated and integrated sensing capabilities in soft robotic designs.

## Supporting information

**S1 Video. Example of breaking a specimen by hand before mechanical testing.**
(MP4)

**S2 Video. Blocking force test for the two compared bellow-jointed fingers.**
(MP4)

**S3 Video. Actuating individual bellows in the fingers.** Each joint of the finger is actuated to their respective extents and three samples of delicate grasping is shown.
(MP4)

**S4 Video. Pick-and-place task of various objects done with the example three-finger gripper.** 10× sped up footage of the three-finger gripper prototype mounted on the end of WAM Arm grasping various objects and dropping them in a bin.
(MP4)

**S1 Fig.**
(JPG)

## Acknowledgments

We'd like to thank Brian Strzelecki and Matt Kindig at the Center for Limb Loss and Mobility (CLiMB) for their help with 3D printing and material testing, and David Caballero for his help with the Barrett WAM Arm. We also thank members of the Rombolabs for their input.

## Author Contributions

**Conceptualization:** Pornthep Preechayasomboon, Eric Rombokas.

**Data curation:** Pornthep Preechayasomboon.

**Formal analysis:** Pornthep Preechayasomboon.

**Investigation:** Pornthep Preechayasomboon.

**Methodology:** Pornthep Preechayasomboon.

**Project administration:** Pornthep Preechayasomboon, Eric Rombokas.

**Resources:** Pornthep Preechayasomboon, Eric Rombokas.

**Software:** Pornthep Preechayasomboon.

**Supervision:** Eric Rombokas.

**Validation:** Pornthep Preechayasomboon.

**Visualization:** Pornthep Preechayasomboon.

**Writing – original draft:** Pornthep Preechayasomboon, Eric Rombokas.

**Writing – review & editing:** Pornthep Preechayasomboon, Eric Rombokas.

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
