## [Decision Letter · Decision Letter 0]

22 Apr 2020

PONE-D-20-05874

Negshell casting: 3D-printed structured and sacrificial cores for soft robot fabrication

PLOS ONE

Dear Mr. Preechayasomboon,

Thank you for submitting your manuscript to PLOS ONE. After careful consideration, we feel that it has merit but does not fully meet PLOS ONE’s publication criteria as it currently stands. Therefore, we invite you to submit a revised version of the manuscript that addresses the points raised during the review process.

We would appreciate receiving your revised manuscript by May 30 2020 11:59PM. To enhance the reproducibility of your results, we recommend that if applicable you deposit your laboratory protocols in protocols.io, where a protocol can be assigned its own identifier (DOI) such that it can be cited independently in the future. For instructions see: http://journals.plos.org/plosone/s/submission-guidelines#loc-laboratory-protocols

We look forward to receiving your revised manuscript.

Kind regards,

Ning Cai, Ph.D.

Academic Editor

PLOS ONE

Journal Requirements:

2. We note that Figures in your submission contain copyrighted images. All PLOS content is published under the Creative Commons Attribution License (CC BY 4.0), which means that the manuscript, images, and Supporting Information files will be freely available online, and any third party is permitted to access, download, copy, distribute, and use these materials in any way, even commercially, with proper attribution. For more information, see our copyright guidelines: http://journals.plos.org/plosone/s/licenses-and-copyright.

a)    You may seek permission from the original copyright holder of Figures to publish the content specifically under the CC BY 4.0 license.

Reviewers' comments:

Reviewer's Responses to Questions

**Comments to the Author**

1. Is the manuscript technically sound, and do the data support the conclusions?

Reviewer #1: Yes

Reviewer #2: Yes

2. Has the statistical analysis been performed appropriately and rigorously? 

Reviewer #1: Yes

Reviewer #2: N/A

3. Have the authors made all data underlying the findings in their manuscript fully available?

Reviewer #1: Yes

Reviewer #2: Yes

4. Is the manuscript presented in an intelligible fashion and written in standard English?

Reviewer #1: Yes

Reviewer #2: Yes

5. Review Comments to the Author

Reviewer #1: The authors presented a novel fabrication method for soft robots. In this paper, the negshell casting was used in the bellow-jointed finger. The experimental data fit well with the original design idea. However, the reviewer still have some comments:

1. The fragment of the negshell may do harm to the soft material, and reduce the life of the robot.

2. Also, the fragment of the negshell may block the inner channel of the robot.

3. In my opinoin, the authors should consider the circle life of the actuator due to the questions I mentioned above.

Reviewer #2: The manuscript, "Negshell casting: 3D-printed structured and sacrificial cores for soft robot fabrication" by Preechayasomboon and Romobokas, presents an interesting idea that can be employed in fabricating 3D-printed structures, especially for soft robotics. When fabricating a hollow 3D printed structure using stereolithography (SLA) based methods (and, in fact, with fused deposition model (FDM) based methods as well), users face problems of dealing with unwanted auxiliary structures. When a 'hollow' space is programmed, such space is usually filled with artifacts with some geometric features that the printing algorithm determines. Such filled artifacts are often nuisance. The authors of this manuscript made a contrarian perspective to design 'sacrificial artifact' to fill the internal volume when 3D-printing such hollow structure, and named this fabrication method as "negshell casting", as an acronym for negatice-space eggshell casting.

I rate this idea to be extremely practical and useful. Although this is not a profound science, such simple idea can really make contribution to wide fields of science and engineering that utilizes 3D printing as a tool. The authors included details in design and fabrication, and included an interesting application example of a pneumatic gripper that can perform pick and place. The detailed writing style of the manuscript is consistent with the purpose of PLOS ONE as an open-source of ideas. I would like to recommend a publication of this paper with the current form.

---

## [Author Response · Author response to Decision Letter 0]

20 May 2020

Dear Dr.Ning Cai & Reviewers,

First of all, we would like to thank the editors and all reviewers for their valuable input for our paper. We present here a revised copy of our manuscript. We would like to point out issues raised in the original review and letter with the following:

1.To enhance the reproducibility of your results, we recommend that if applicable you deposit your laboratory protocols in protocols.io

Response: We have opted to use GitHub as our repository for the project instead as protocols.io requires a paid plan for projects larger than 100MB and has a strict layout, which is not ideal for engineering projects. We host our data and protocols on GitHub Pages instead: https://negshell.github.io/, this gives us much more flexibility in terms of presentation and content.

2. Please ensure that your manuscript meets PLOS ONE's style requirements

Response: We have read through our manuscript carefully to ensure that it complies to the PLOS One formatting guidelines and have touched up any mistakes found.

3.We note that Figures in your submission contain copyrighted images

Response: We were unsure which figure had copyrighted images, as all figures were generated by ourselves. Figure 17 is the only image that has possibly copyrighted material: the logo of a whiteboard marker company. We have edited the Figure to not contain the logo. Any other logo (Rombolabs, VA, CLiMB, etc. are our own and we have the right to use them along with our affiliation with the respective groups)

4. From Reviewer 1:

"1. The fragment of the negshell may do harm to the soft material, and reduce the life of the robot.

2. Also, the fragment of the negshell may block the inner channel of the robot.

3. In my opinoin, the authors should consider the circle life of the actuator due to the questions I mentioned above."

Response: Reviewer 1 has brought up a good point about the longevity of negshell-produced soft robots. We agree that the fragments can cause issues if cross-hatch is designed improperly. Blockage can be mitigated by making the pattern smaller than the smallest orifice in the design thus even if a fragment is lodged in an orifice, the fragment will have perforations to allow fluids to travel through. As for the sharpness of the fragments, we argue that as the fragments are created while physically crushing the negshell cores during the fabrication process, the soft material used must be able to mechanically withstand them anyways. Our fabrication process would be invalidated if they were in fact causing damage to the structure, which our prototypes and experiments show otherwise. Furthermore, we have found that the 0.4 mm thickness negshell cores are in fact soft and compliant themselves, though this is not covered in the paper as we did not rigorously characterize them. We have addressed these issues in our revised manuscript in the Discussion section. 

We hope that the revisions satisfy the reviewers concerns, and are welcome to any additional comments. 

Thank you & Best regards,

Pornthep Preechayasomboon

Eric Rombokas

---

## [Editor Report · Decision Letter 1]

26 May 2020

Negshell casting: 3D-printed structured and sacrificial cores for soft robot fabrication

PONE-D-20-05874R1

Dear Dr. Preechayasomboon,

We are pleased to inform you that your manuscript has been judged scientifically suitable for publication and will be formally accepted for publication once it complies with all outstanding technical requirements.

With kind regards,

Ning Cai, Ph.D.

Academic Editor

PLOS ONE
---

## [Editor Report · Acceptance letter]

3 Jun 2020

PONE-D-20-05874R1 

Negshell casting: 3D-printed structured and sacrificial cores for soft robot fabrication 

Dear Dr. Preechayasomboon:

I'm pleased to inform you that your manuscript has been deemed suitable for publication in PLOS ONE. Congratulations! Your manuscript is now with our production department. 

Kind regards, 

on behalf of

Dr. Ning Cai 

Academic Editor

PLOS ONE